# The Role of the Transforming Growth Factor-β Signaling Pathway in Gastrointestinal Cancers

**DOI:** 10.3390/biom13101551

**Published:** 2023-10-19

**Authors:** Tasuku Matsuoka, Masakazu Yashiro

**Affiliations:** Molecular Oncology and Therapeutics, Osaka Metropolitan University Graduate School of Medicine, Osaka 5458585, Japan; t22738q@omu.ac.jp

**Keywords:** transforming growth factor-β, smad, gastrointestinal cancers, epithelial-mesenchymal transition, tumor microenvironment, immune evasion, targeted therapy

## Abstract

Transforming growth factor-β (TGF-β) has attracted attention as a tumor suppressor because of its potent growth-suppressive effect on epithelial cells. Dysregulation of the TGF-β signaling pathway is considered to be one of the key factors in carcinogenesis, and genetic alterations affecting TGF-β signaling are extraordinarily common in cancers of the gastrointestinal system, such as hereditary nonpolyposis colon cancer and pancreatic cancer. Accumulating evidence suggests that TGF-β is produced from various types of cells in the tumor microenvironment and mediates extracellular matrix deposition, tumor angiogenesis, the formation of CAFs, and suppression of the anti-tumor immune reaction. It is also being considered as a factor that promotes the malignant transformation of cancer, particularly the invasion and metastasis of cancer cells, including epithelial-mesenchymal transition. Therefore, elucidating the role of TGF-β signaling in carcinogenesis, cancer invasion, and metastasis will provide novel basic insight for diagnosis and prognosis and the development of new molecularly targeted therapies for gastrointestinal cancers. In this review, we outline an overview of the complex mechanisms and functions of TGF-β signaling. Furthermore, we discuss the therapeutic potentials of targeting the TGF-β signaling pathway for gastrointestinal cancer treatment and discuss the remaining challenges and future perspectives on targeting this pathway.

## 1. Introduction

Transforming growth factor-β (TGF-β) signaling is essential for the homeostasis of epithelial cells, stromal fibroblasts, and immune cells. A variety of cells produce and respond to TGF-β, which results in a complicated network that is closely related to tumor development during each process. Thus far, it has been accepted that TGF-β can have different, even paradoxical, roles according to cancer cell types and/or the tumor microenvironment (TME) [1]. In normal and precancerous cells, TGF-β mostly functions as a tumor suppressor by impairing cell growth, facilitating apoptosis, and sustaining genome stability [1]. In early cancer stages, TGF-β signaling represses the growth of tumor cells by inducing apoptosis. In contrast, in late-stage cancers, TGF-β is upregulated and converted into a central driver for tumor development and metastasis. The diverse activity of TGF-β signaling provides cancer-specific properties. Approximately 40% of genetic alterations in mediators of TGF-β signaling occur in gastrointestinal (GI) cancers [2,3,4]. Cancer cells that maintain the capacity of TGF-β signaling capitalize on platforms that induce plasticity, such as epithelial-to-mesenchymal transition (EMT) and stem-like self-renewal, which lead to composition and remodeling of the TME and immune evasion. These multifunctions regulate the progression of cancer cells, which is acknowledged as the “TGF-β paradox”, and have attracted great attention as a potential therapeutic target for GI cancers.

In this review, we summarize the current understanding of the various activations of TGF-β and its signaling pathways. We then highlight the recent progress and challenges in targeting TGF-β signaling and discuss the perspectives of TGF-β-targeting therapeutics that would be important in providing benefits from GI cancer therapies.

## 2. TGF-β Cell Biology

TGF-β generally exists as homodimers or heterodimers in the extracellular region. In the late 1970s, murine sarcoma virus-transformed cells were found to produce numerous growth factors, including TGF-β. TGF-β was discovered as a cytokine that induces normal fibrocyte transformation and anchorage-independent proliferation of fibroblast cell lines [5]. In recent decades, TGF-β has been generally acknowledged to be a multifunctional cytokine that regulates numerous important events in cancer cell activities, such as cell cycle progression, differentiation, adhesion, apoptosis, and immune tolerance [6,7]. Malignant cells can impair TGF-β-induced proliferative suppression by acquiring stimulating mutations, such as those of RAS, and mutations in tumor suppressor genes, such as p53 and retinoblastoma protein. TGF-β enhances forkhead box P3 gene expression [8] and reduces the gene expression of MHC class I-chain-related molecule A, interferon-gamma (IFNγ), natural cytotoxicity receptor 3 granzyme A/B, and perforin to repress immune activity [9,10,11]. In contrast, it increases the expression of interleukin (IL)-11 and metalloproteinase 9 to promote metastasis to certain organs [12,13]. TGF-β also facilitates angiogenesis by enhancing vascular endothelial growth factor (VEGF), connective tissue growth factor, and metalloproteinase 2 levels [14,15].

To elucidate TGF-β activity and targeting, it should be known that TGF-β is synthesized in the endoplasmic reticulum as part of a latent complex in the extracellular matrix (ECM) with no activity (Figure 1). It comprises a signal peptide in the large N-terminal protein named latency-associated peptide (LAP) and a short C-terminal mature peptide [16,17]. When the N-terminal signal peptide is removed, the pro-peptide accumulates into a disulfide-linked homodimer, and latent TGF-β is transformed into homodimeric ligands [18]. The large latent complex (LLC) consists of TGF-β, LAP, and latent TGF-β binding protein (LTBP). LTBP contributes to extracellular release, localization, and activation of latent TGF-β. Latent TGF-β is collected in the ECM and transforms into activated TGF-β under enzymatic and/or non-enzymatic activity. Activated TGF-β can only connect to the TGF receptor complex and then activate canonical and noncanonical TGF-β signal transduction. Latent TGF-β is activated by binding to integrins β1, β6, and β8 via the RGD (Arg Gly-Asp) sequence present in LAP. Meanwhile, LTBP binds to the ECM and is activated via mechanical tension. Integrins, such as αvβ1, αvβ3, αvβ5, αvβ6, and αvβ8, are the main activators of TGF-β ligand. Integrin family members have important roles in the detection and activation of TGF-β [19,20,21]. TGF-β function mediated by integrins (integrins αvβ6 and αvβ8) is crucial for tumorigenesis and immunity. Both αvβ6 and αvβ8 regulate TGF-β signaling via the RGD sequence present in LAP, depending on actin cytoskeleton-generated tensile strength [22]. In numerous cell types, TGF-β is secreted as a complex bound to LTBP and released outside the cells. Meanwhile, in mesenchymal stromal cells, regulatory T cells (Tregs), and vascular endothelial cells, TGF-β remains on the cell membrane surface-formed complex bound to glycoprotein-A repetitions predominant (GARP) via disulfide bonds. When TGF-β is stimulated by integrin αVβ8, it can activate adjacent cells, such as Tregs. In addition, LRRC33, a GARP-related molecule, is expressed in cells of the central nervous system [23]. LRRC33 is relevant to latent TGF-β and mediates TGF-β activity. A human neutralizing antibody targeting GARP has been developed and is expected to specifically regulate the immunosuppressive effects of Tregs.

Nowadays, it is considered that the functions of TGF-β are mainly divided into four categories: (1) inhibition of cell proliferation; (2) promotion of EMT; (3) interaction with TME; and (4) immunosuppression. TGF-β was discovered to suppress the growth of many cells, including epithelial cells, blood cells, lymphocytes, and vascular endothelial cells, in the mid-1980s, and its tumor-suppressing effects have attracted attention in relation to cancer for a long time.

Loss of the TGF-β tumor suppressive response plays an important role in developing human cancers. As a central player in TGF-β signal transduction, small mothers against decapentaplegic (SMAD) 4, which is also known as DPC4, which was detected as a tumor suppressor gene for pancreatic cancer (PC), is frequently mutated or deleted in GI cancer. On the contrary, in the mid-1990s, it was demonstrated that TGF-β promotes EMT, indicating that TGF-β enhances late-stage cancer progression [24].

TGF-β consists of a large group of organizationally linked cell regulatory proteins, which is called the TGF-β superfamily. The discovery of TGF-β family members and their signaling components has provided an understanding of the complicated biological functions of TGF-β. The TGF-β family comprises 33 members in mammals, such as TGF-β isoforms and activin, nodal and bone morphogenetic proteins (BMPs), and TGF-β family signaling contained in a variety of physiological processes. It is also associated with cancer development, including proliferation, differentiation, and migration [25]. TGF-β has three structurally similar mammalian genome-encoded isoforms, namely, TGF-β1, TGF-β2, and TGF-β3 [25,26]. All three isoforms have been said to elicit equivalent biological activities but also have different activated mechanisms [26]. According to the database from The Cancer Genome Atlas (TCGA), the TGF-β1 homodimer is the most widely examined isoform and is expressed in various kinds of human cancer. Moreover, the expression of TGF-β1 is the most closely associated with the TGF-β pathway compared with TGF-β2, and 3 [26]. Although other TGF-β superfamily members have similarly important roles in the regulation of biological procedures, they are not described much in this review.

## 3. TGF-β Signaling Pathways

### 3.1. The Receptor of the TGF-β Family

TGF-β signals are transmitted by the complex TGF-β receptor composed of TGF-β receptor type 1 (TGFBR1, also called ALK-5) and receptor type 2 (TGFBR2). Both TGFBR1 and TGFBR2 have two paired serine-threonine kinase domains that phosphorylate serine and threonine in the cell. TGF-β binding to a ligand induces the construction of tetrameric receptor complexes containing TGFBR1 dimers and TGFBR2 dimers, which phosphorylate TGFBR1 on a specific Gly/Ser-rich “GS sequence (glycine-serine) region and result in transmitting signals to the intracellular region [25] (Figure 1). Other members of the TGF-β family either bind to their own receptors or compete with other TGF-β family receptors to transduce and regulate signals. Activin mainly binds and activates type 1 receptors called ALK-4 and ALK-7, while BMP mainly activates ALK-1, -2, -3, and -6.

### 3.2. Smad Signaling Pathways

Although there are many TGF-β ligands, fewer receptors and downstream effectors, such as SMAD proteins, regulate intracellular signaling. A hetero-tetrameric complex of TGFBR2 and TGFBR1 results in the TGFBR2 kinase domain phosphorylating TGFBR1 and subsequently activating SMAD2/3 [25]. Consequently, activated SMAD2/3 facilitates complexes connecting to SMAD4, generating a trimeric complex that is then translocated into the nucleus, where they can exhibit and inhibit target gene expressions (Figure 1) [27]. Downstream factors of the SMAD signal pathway, particularly Smad2/Smad3, are considered to be important regulators of the TGF-β pathway in tissue fibrosis and tumorigenesis. In contrast, TGF-β produced SMAD6/SMAD7 to serve as a negative mediator of TGF-β/SMAD signals, which is relevant to TGFBR1, leading to inhibition of the activation of SMAD2/3. Furthermore, SMAD7 provokes TGF-β signals, thereby disrupting the structure of the SMAD-DNA conjugate in the nucleus [28] and blocking the translocation of the SMAD2/SMAD4 conjugate [29].

TGF-β can also introduce SMAD-independent noncanonical signaling pathways as well as the canonical SMAD pathway (Figure 1) [30]. TGFBR1 stimulates RHO small GTPases, which mediate the function of RHO-associated protein kinase and LIM kinase, thereby remodeling the actin cytoskeleton [31]. Similarly, TGFBR2 activates the cell polarity regulator PAR6, which mediates tight junctions and the migration/motility of cancer cells [31]. TGF-β also directly phosphorylates TGF-β-activated kinase 1 (TAK1), which results in prompt c-Jun NH2-terminal kinase (JNK), p38 mitogen-activated protein kinase (MAPK), and nuclear factor kappa-B (NF-κB) pathways. The PI3K/AKT/mTOR pathway is also induced as the downstream signaling pathway for noncanonical signals [31]. Furthermore, TGF-β activates Src homology domain 2-containing protein (SHC) and then stimulates growth factor receptor-binding protein 2 (GRB2) and son of sevenless (SOS), which leads to initiating the RAS, RAF, extracellular signal-regulated kinase (ERK), and MEK pathways [31]. The RAS-responsive element-binding protein 1 (RREB1) serves as a connection between the RAS and TGF-β pathways, which participate in the synchronized induction of a fibrogenic EMT pathway [32]. There is crosstalk between the canonical and noncanonical TGF-β pathways, and it is controlled by tyrosine kinase receptors [33]. Moreover, these pathways are also stimulated by TGF-β through inducing platelet-derived growth factor in an autocrine or paracrine manner [34].

## 4. Role of TGF-β Signaling in the Carcinogenesis and Progression of GI Cancer

TCGA dataset analysis shows that genetic alterations in TGF-β signaling are represented in about 30% of GI tumor specimens; in particular, mutations of TGFBR2 and SMAD4 are common in PC and colorectal cancers (CRCs) (Table 1) [2]. In PC, novel evidence suggests that several mutations in the TGF-β signaling genes, such as TGFBR1, TGFBR2, SMAD2, and SMAD4 genes, occur [35]. Notably, SMAD deletion or mutation is identified in 60% of PC patients, and an increased SMAD mutation drives early carcinogenesis and dissemination [36]. In CRC, SMAD4 is one of the most frequently mutated genes based on a series of high-throughput analyses. A recent study utilizing next-generation sequencing (NGS) including 123 non-MSI-high mCRC patients demonstrated an about 20% mutation occurrence of SMAD4 [37]. TGFBR2 mutations are also often shown in CRC [38]. Remarkably, TGFBR2 mutations are shown in 80–90% of CRC with microsatellite instability [39]. Carcinoembryonic antigen-associated cell adhesion molecule 5 (CEACAM5), a highly glycosylated protein of the CEACAM family, reduces the expression of TGF-β pathway members (TGFBR2, SMAD4, and SPTBN1), which alters the colonic microbiome to promote CRC [40]. By contrast, SMAD2 mutations have been reported in hepatocellular carcinoma (HCC) [41]. For SMAD4, about 70% of esophageal adenocarcinoma (EAC) associated with Barrett’s esophagus (BE) revealed loss of heterozygosity in a region involving SMAD2 and SMAD4 on chromosome 18q [42,43]. This loss of heterozygosity may be an early event in neoplastic transformation because it occurs in approximately 30–70% of patients with precancerous BE [43]. Another mechanism for reducing SMAD4 expression is the methylation of the SMAD4 promoter in both BE and EAC [44]. Genomic alterations of SMAD4 are more frequent in EAC (24%) than in esophageal squamous cell carcinoma (ECC) (8%). The noncanonical TGF-β and JNK signaling axis was hyperactivated in EC patients, and the genes mediated by these pathways were also upregulated in EC, suggesting new treatment tools for this refractory cancer [45]. The frequency of genetic alterations in TGF-β pathway-related genes in GI cancers was summarized in Table 1.

Since the TGF-β signaling pathway has diverse tumorigenic effects, understanding its mechanisms in carcinogenesis will lead to better diagnostic and predictive tools as well as therapeutic approaches.

## 5. Role of TGF-β Signaling in Epithelial-Mesenchymal Transition

EMT is a phenomenon in which keratinocytes differentiate into mesenchymal cells and contribute to the initiation processes, migration, and invasion of cancer cells [58]. During EMT, the down-regulation of epithelial markers such as E-cadherin, occludins, and claudins is shown, whereas mesenchymal markers such as N-cadherin, vimentin, and fibronectin are up-regulated. These alterations result in reduced cell adhesion, loss of polarity, tight junctions, and a highly invasive mesenchymal phenotype. TGF-β has a potent role in initiating tumorigenesis and metastatic ability of cancer cells by inducing the EMT via the activation of transcription factors such as SLUG, SNAIL, ZEB1/2, DMNT, and TWIST (Figure 2) [59]. TGF-β can also induce EMT in non-canonical signaling, for instance, by fascinating cytoskeletal remodeling, which results in the phosphorylation of ERK [60]. MET (mesenchymal-epithelial transition) is the reverse process of EMT. EMT not only transforms epithelial cells into full-form mesenchymal cells (full EMT) but often also partially acquires the characteristics of interstitial cells through intermediate hybrid states (partial EMT state, P-EMT). Furthermore, cells that have undergone EMT may acquire epithelial-like cell phenotypes due to MET. TGF-β vigorously regulates the transition between P-EMT and full-EMT positions in cancer cells [61]. Furthermore, long-term TGF-β stimulation, as often seen during clinical cancer progression, can stabilize EMT, in contrast to the reversible EMT induced by short-term TGF-β stimulation. Long-term TGF-β stimulation activates not only the SMAD pathway but also mTOR signaling, which stabilizes cancer stem cell (CSC)-like properties and anticancer drug resistance [62].

Accordingly, TGF-β is closely related to the infiltration and metastasis of GI cancer through EMT. In this section, we review the relationship between TGF-β and EMT in GI cancer. For example, in PC, it is conceivable that TGF-β induces EMT from an early stage due to KRAS abnormalities and shifts from a tumor suppressor to a tumor promoter [63]. Alterations in signaling or gene expression that facilitate EMT promote the CSC population and, consequently, the tumor-initiating capacity. The inhibitor of DNA binding 1 (ID1) protein acts as a cell differentiation suppressor and growth stimulator. Transcriptional dysregulation of ID1 regulates the escape of pancreatic ductal adenocarcinoma (PDA) from TGF-β tumor suppression and decouples TGF-β-induced EMT from apoptosis, which contributes progenitor-like characteristics to PDA cells [64]. In EC, cell division cycle associated 7, one of the copy number amplification genes, was shown to facilitate the EMT process and metastasis of EC through TGF-β signaling activation via inhibiting SMAD7 mRNA transcription and enhancing SMAD4 mRNA transcription [65]. Epithelium-specific ETS transcription factor 1 (ESE1) belongs to the ETS-domain transcription factor superfamily. In PC, an ESE1/anterior gradient protein 2 axis negatively interacts with TGF-β signaling to mediate the EMT phenotype by functioning with EMT drivers such as ZEB1/2 [66]. In addition, TGF-β1 upregulates acyl-CoA synthetases 3, leading to promoting ATP and suppressing NADPH, thereby maintaining redox homeostasis and facilitating the EMT and metastasis of CRC cells [67].

## 6. Role of TGF-β Signaling in the Tumor Microenvironment

During cancer development, the surrounding components in TME co-exist with cancer cells via paracrine and autocrine mechanisms. These active signaling networks in the TME confer cancer advancement, metastasis, and chemoresistance [68]. Hence, the TME is predicted to be useful as a tool for cancer therapy because of the enlarged and precise acknowledgement of the heterogeneity and complexity of cancer cells. The TME is organized into a variety of cell types, such as cancer-associated fibroblasts (CAFs), immune-related cells, endothelial cells, adipocytes, stromal cells, and the ECM, as well as a wide range of cytokines [69,70]. TGF-β is a potent cytokine that enables it to mediate the activity of most cells existing in the TME, which results in producing ECM components, stimulating CAFs, and facilitating angiogenesis (Figure 3).

Inflammation is a biological defensive reaction to a variety of damaging stimuli, including injured cells, pathogens, and toxic factors [71]. Chronic inflammation facilitates numerous pathological situations, such as tissue fibrosis [72]. The fibrotic procedure is characterized by altered mesenchymal transition and the growth of fibroblasts. One of the features of fibrosis is the transformation of fibroblasts into myofibroblasts, which express α-smooth muscle actin (α-SMA) and dysregulate the ECM. TGF-β has been shown to be crucial for the initiation of the fibrotic response and for the activity of the stroma surrounding cancer cells. TGF-β directly stimulates target cells to synthesize and transport ECM components such as fibronectin, laminins, and collagens [73]. TGF-β stimulation induces HCC cells to release an excess of soluble factors, which lead to target stromal cells. Connective tissue growth factor (CTGF), which is produced by TGF-β-stimulated HCC cells, promotes fibrosis, which results in enhancing tumor size in the in vivo model of HCC [74]. In scirrhous GC, TGF-β secreted from gastric fibroblasts affects the invasion ability of EMT and, in turn, prompts peritoneal metastasis [75].

As key components of the TME, CAFs are initiated by the activation or transformation of precursor cells in tumor tissues [76]. Activated CAFs provide a favorable environment to promote ECM remodeling, cancer cell growth, invasion, angiogenesis, metabolic reprogramming, stemness, and tumor metastasis. TGF-β signaling is a pivotal trigger of CAF function and construction and contributes to maintaining the morphological features and functional phenotypes of CAFs. In PCs, CAFs are identified as distant from primary tumors with low α-SMA expression and high levels of IL6, and myofibroblastic CAFs are located adjacent to tumor sites with high α-SMA expression and demonstrate a robust TGF-β reaction [77]. In stroma-abundant PC, branched chain α-ketoacid (BCKA) was positively related to the catabolism initiated by the metabolic reprogramming of CAFs and PC cells [78]. The TGF-β/SMAD5 axis induced the incorporation of ECM, which provided the amino acid precursors in CAFs, leading to BCKA secretion [78]. In HCC, CAFs stimulated the construction of vascular mimicry, which was significantly diminished when TGF-β signaling was eliminated [79]. In GC, several studies have demonstrated the relationship between TGF-β and stemness. In a recent study, fibroblasts activated by Helicobacter pylori promoted the differentiation of gastric epithelial cells, which is associated with CSCs. This result is partially dependent on TGF-β signaling and promotes tumorigenesis [80]. CAFs have been found to facilitate the aggressive phenotypes of scirrhous GC cells, which involve adhesion to mesothelial cells [81], migration and invasion [82], peritoneal dissemination [83], and the CSC phenotype [84]. Using a comprehensive study on the basis of large-scale clinical samples, including 18 datasets of 2320 patients with CRC from the TCGA and GEO databases, a recent study established the CAF-derived long non-coding RNA (CAFDL) signature in the TCGA-CRC training set, evaluated its predictive significance, and demonstrated that CAFDL was significantly positively associated with TGF-β1 [85]. TGF-β was shown to suppress lymphangiogenesis via inhibiting the expression of collagen and calcium-binding EGF domain-1 [86], a molecule implicated in the proteolytic activity and maturation of VEGF-C [87], showing a new mechanism by which a deficit of TGF-β signaling facilitates metastasis of CRC [86]. Accurate and effective therapy targeting the TGF-β pathway in CAFs is an assuring tool for GI cancers, and increasing attempts are ongoing to attain this purpose.

## 7. Role of TGF-β Signaling in the Immune System

TGF-β is also known as a crucial immunosuppressive cytokine [88]. Immunosuppression by TGF-β involves various immune cells, such as T cells, B cells, tumor-associated macrophages (TAMs), tumor-associated neutrophils, dendric cells (DCs), NK cells, and myeloid-derived suppressor cells (MDSCs) (Figure 3). TGF-β prompts macrophages to shift from an inflammatory (M1) to a tumor-trophic (M2) phenotype to be transformed into TAM, thus producing an immunosuppressive microenvironment [89,90]. In a similar way, TGF-β produces an N2 neutrophil phenotype, which facilitates cancer development [91]. Furthermore, up-regulation of TGF-β promotes immunosuppressive activity by translating naïve T cells to Treg cells [92]. TGF-β can inhibit the activation and cytotoxicity of NK cells and suppress their IFN-γ secretion [7]. IFN-γ is an activator of macrophages and promotes NK cells and neutrophils. As well, TGF-β represses the development of DCs and helper T cells [92]. Moreover, TGF-β can mediate the activation, proliferation, and apoptosis of B cells and regulate antibody switching in B cells [93]. Meanwhile, TGF-β can induce the expression of forkhead box P3, which leads to the production and differentiation of Treg from CD4+ T cells through IL-2 signaling. It also prompts differentiation into IL-17-producing helper T cells (Th17 cells) in the presence of IL-6 and IL-21 [92,94]. TGF-β produced by Tregs suppresses the differentiation and growth of effector T cells, has a role in inhibiting the function and development of cytotoxic CD8+ T cells by repressing the tumor antigen processing and presentation of DCs, and inhibits CD8+ T cell proliferation through blocking the expression of IFN-γ and IL2 [95,96,97]. Thus, TGF-β is known to regulate T cell differentiation depending on the existence of inflammatory cytokines. The blockade of the TGF-β pathway is an efficient tool in the treatment of cancer, which attenuates Treg-mediated anti-tumor activity, enhances T cell cytotoxicity, and prompts T cell movement into the center of the tumor, leading to inducing robust anti-tumor immunity and tumor suppression (Figure 3) [98]. In CRC, macrophage polarization to M2 phenotypes of TAMs via the TGF-β pathway was induced by the release of Collagen Triple Helix Repeat Containing 1 (CTHRC1), which led to liver metastasis in vivo models [99]. A previous study described that CD4^+^ CD25^+^ Tregs-derived IL-10 has been shown to inhibit microbially-induced colonic inflammation and CRC development [100]. In PC, TAMs promote migration and invasion of cancer cells by inducing EMT via the TGF-β-SMAD2/3/4-Snail signaling axis [101]. In EC patients, the effector immune responses correlated with the CD3+CD8+ T lymphocytes are disordered. CD8+ T lymphocyte repressed function could be associated with aberrant miR-21 production, which could be potentially involved in a pivotal target for immunotherapeutic strategies [102]. Collectively, these findings further imply the complications of the TGF-β pathway in the immune evasion of GI cancer and provide a pivotal role for immune cell modulation in the tumor-facilitating efficiencies of the TGF-β signaling pathway in GI cancer.

## 8. Role of Non-Coding RNAs Involved in TGF-β Signaling

The majority of RNA-transcribed but not-encoded proteins can be categorized as non-coding RNAs (ncRNAs) [103]. Depending on the shape, length, and situation, ncRNAs have been subdivided into diverse classes, such as microRNA (miRNA), long ncRNA (lncRNA), circular RNA (circRNA), and PIWI-interacting RNA (piRNA). They can modulate transcription factors to bind to promoters and thus regulate the progression of a variety of malignant tumors [103]. Growing evidence has shown that TGF-β signaling in tumor progression is related to the alteration of ncRNA regulation. In CRC, LINC00941 has tumorigenic functions and is highly expressed. Moreover, it facilitates the metastasis of CRC by activating the TGF-β/SMAD2/3 signaling pathway and preventing SMAD4 protein degradation [104]. LncRNA SBF2-AS1 regulates TGFBR1-mediated signaling by sponging miR-140-5p, which facilitates HCC progression [105]. LncRNA MBNL2-AS1, which forms a competing endogenous RNA network with miR-424-5p and SMAD7, downregulated the TGF-β/EMT pathway, thereby inhibiting the invasion ability of gastric cancer (GC) cells [106]. Circ GFRA1 (hsa_circ_005239), which originated from the GFRA1 (GDNF family receptor alpha1) on chromosome 10, is a type of cancer-related circRNA. The up-regulation of circGFRA1 significantly prompts tumor growth and intra-hepatic metastases in HCC. Meanwhile, similar binding sequences were identified between circGFRA1 and miR-498, and these expressions are reversely associated in HCC cells [107].

Abundant studies suggest that ncRNAs are mediated in response to TGF-β-induced EMT in GI cancer. For example, lncRNAs can mediate EMT advancement by phosphorylating TGF-β signaling in PC. For instance, lnc00462 facilitates PC invasion and metastasis via the miR-665/TGFBR1-TGFBR2/SMAD2/3 pathway [108]. As well, miR-492 induces EMT by stimulating TGF-β/SMAD3 signaling and the expression of NR2C1, a member of the steroid/thyroid hormone receptor superfamily, suggesting the significance of miR-492-based treatments for PC [109]. A recent paper described that the activation of CXCL12/CXCR7 promoted miR-146a-5p and miR-155-5p, derived from the exosome in CRC cells, to enhance the activation of CAFs via the JAK2-STAT2/NF-κB signaling pathway and further promote EMT and CRC metastasis to the lung [110]. Macrophage-derived exosomal miR-501-3p was shown to facilitate PC metastasis through the suppression of TGFBR3, which results in the activation of the TGF-β signaling pathway [111]. The interaction of ncRNAs and TGF-β signaling may provide a latent new target and promising diagnostic and prognostic markers for GI cancer. Table 2 summarizes non-coding RNAs involved in TGF-β signaling in GI cancers.

## 9. Discussion and Future Perspective

According to the World Health Organization, approximately 5.5 million new cases of GI cancer worldwide have been reported in 2020 [175]. Over 14 million new cancer cases are diagnosed annually, with 8.2 million deaths. Given the high mortality rates, the capacity to overcome GI cancer is urgently needed [176]. Many therapy strategies are available for patients with GI cancer, such as surgery, chemotherapy, and chemoradiotherapy [177]. However, 5-year overall survival (OS) remains unsatisfactory for patients with advanced stages, and effective methods to evaluate the prognosis of patients with GI cancer are still lacking [178]. Understanding the molecular mechanisms controlling the heterogeneity of GI cancer could lead to new therapeutic targets. We have evidence that relates TGF-β signaling to cancer biology, from tumorigenesis to TME and immune evasion in GI cancer. The importance of TGF-β signaling in GI cancer is represented by the high incidence of genetic alterations in the TGF-β pathways. Due to its pleiotropic activity as a tumor suppressor and active promoter of tumor development and metastasis, targeting TGF-β could provide new and promising prospects in GI cancer treatment.

Therapeutic strategies have normally been distinguished into three types: ligand, ligand-receptor binding, and intracellular transduction to disrupt TGF-β signaling. These strategies include the use of neutralizing antibodies, TGF-β ligand traps, small-molecule TGF-β inhibitors, antisense oligonucleotides, and vaccine-based therapy. Thus far, non-coding RNAs, kinase inhibitors, and natural compounds are familiar approaches that have been used for targeting TGF-β in clinical trials (Table 3). The remarkable capacities of anti-tumor agents with TGF-β-neutralizing antibodies and selective small-molecule TGF-β receptor kinase inhibitors have been proven in recent decades. In contrast, although TGF-β-targeted therapies have been efficacious in in vitro and in vivo examinations, the application of these pharmacological agents for approved subsequent clinical trials was unsatisfactory because of their systemic efficacy on normal tissues of TGF-β, which causes undesirable side effects such as fatigue, gingival bleeding, headache, epistaxis, and TGF-β-related skin lesions, including reversible cutaneous keratoacanthomas/squamous-cell carcinomas [179,180]. In particular, some TGF-β receptor inhibitors and neutralizing antibodies have caused cardiac toxicity in animal models. Although inhibition of TGF-β signaling has meaningful clinical capacity for treating advanced-stage cancers and metastasis, TGF-β blocking alone may promote tumor proliferation [181]. For instance, blocking TGF-β represses ECM deposition and decompresses blood vessels, leading to the recovery of blood perfusion. Enhanced blood perfusion provides a source of nutrients and oxygen to cancer cells and enhances tumor proliferation [181], which could enhance the ability of cancer cells to metastasize. Thus, the use of these inhibitors as monotherapies is still challenging, but combining them with other medications may be considerably more attractive. In PC, a recent phase II clinical trial presented that galunisertib combined with gemcitabine prolonged the outcome of patients at a dose of 300 mg/day [182]. Galunisertib also presented a satisfactory safety profile and improved OS when used in combination with sorafenib, implying its usefulness as a second-line therapy for HCC in a clinical phase II study [183]. In the meantime, in advanced rectal cancer, galunisertib improved the complete response rates to neoadjuvant chemoradiotherapy [184]. Vaccine-based therapy is an emerging new tool for GI cancer therapy. Vigil (Gemogenovatucel-T, FANG™, IND14205) is an autologous compound that consists of a plasmid encoding granulocyte-macrophage colony-stimulating factor and a bifunctional short hairpin RNA targeting furin convertase, which leads to suppressed TGF-β1 and TGF-β2 function. A non-randomized phase I trial evaluating the safety profile of Vigil in HCC patients who are surgically unresectable is now underway (NCT01061840).

Given that the inhibition of the TGF-β pathway induces tumor responses to immune checkpoint inhibitors (ICIs), including programmed death-1 (PD-1) or PD-1 ligand (PD-L1) antibodies, antitumorigenic immunotherapies have been developed to enhance the tumor-inhibitory effect of anti-PD1 treatment by suppressing TGF-β signaling [185]. Furthermore, stemness, EMT, and tumor resistance, which are closely related to TGF-β, are shown to be impaired by dual inhibition of TGF-β/PD-L1 [186]. Therefore, a variety of clinical studies are validating the effect of combining ICIs and TGF-β-targeting agents (Table 2) [92]. For example, the feasibility and usefulness of vactosertib or galunisertib combined with durvalumab, a monoclonal PD-L1 antibody, are under examination in clinical studies of CRC, PC, and GC [185]. Subsequently, M7824, a bispecific fusion peptide against PD-L1/TGF-β, localizes to TME, activates the immune function of Tregs, produces NK and CD8 + T cell activity, and increases the density of anti-tumor type 1 macrophages [187,188]. M7824-treated patients with recurrent gastro-esophageal or GC disease revealed 16% of the objective response rate (ORR) and 26% of the disease control rate. Responses are likely to depend on a high level of TGF-β in the tumor, regardless of the PD-L1 expression profile (NCT02699515) [189]. Monotherapy with SHR-1701, a fusion protein that includes an anti-PD-L1 mAb to which TGFBR2 ORR and 12-month OS of 20% and 54.5%, respectively, among cancers (NCT3710265) [190].

Thus far, no TGF-β-targeting agents have been approved by the FDA for GI cancer therapy. Although the dual blockage of ICIs and TGF-β signaling is well established, several challenges remain in terms of antagonizing TGF-β signaling in tumorigenesis. Limited knowledge of the interaction between proteins comprised in TGF-β secretion, activation, and signal crosstalk is insufficient for utilizing current TGF-β blockades in adjuvant combination therapies [191]. Indeed, the lack of dosing schedules and their dual activity in carcinogenesis make the use of TGF-β antagonists for the treatment of GI cancer challenging [192]. To promote efficient therapies that focus on tumor-promoting functions, screening genetic variants and the TME of each patient would provide great benefits in predicting the effectiveness of TGF-β antagonists. As such, the detection of dependable predictive biomarkers would provide a better strategy for patient choice. Alterations in the TME, such as matrix rigidity and tissue stiffness, can influence the effectiveness of TGF-β [193]. A better understanding of tissue mechanics in the TME may make it possible to provide the cellular response to TGF-β [194]. At present, the most attractive candidates for TGF-β signaling are traditional biomarkers such as SMAD2, SMAD4, and TGFBR2, as shown by several previous studies [195,196,197]. Meanwhile, the identification of blood-based biomarkers is important to select patients who will respond to TGF-β-targeting therapy and to decide the timing of the response of the patients to TGF-β inhibitors to increase therapeutic usefulness. In EC, TGF-β1 levels markedly increased after 40 Gy irradiation in patients who had radiation-induced pneumonitis. In contrast, patients with non-radiation-induced pneumonitis showed steady TGF-β1 levels during and after radiation therapy [198]. To date, clinical studies have assessed the value of the level of TGF-β as a diagnostic and prognostic biomarker in GI cancer (NCT01246986, NCT05500027, NCT04729725, and NCT04122937). Recent studies demonstrate that the secretion of TGF-β in small extracellular vesicles (sEV-TGF-β) contributes to tumor growth, development, and metastasis. Moreover, sEV-TGF-β is also closely related to immune evasion [199]. Interestingly, in non-small-cell lung cancer, the expression of extracellular vesicle (EV) TGF-β was associated with the prediction of efficacy of ICIs and shorter survival, which resulted in a higher accuracy than the existing circulating TGF-β and tissue PD-L1 in patients who are treated with ICIs, suggesting EV TGF-β as a latent predictive biomarker in immunotherapies involving dual TGF-β and PD-L1 blockade [200].

As previously described, reducing toxicity would be a key problem in applying the TGF-β blocking agents for clinical usage [201,202]. In melanoma, specifically blocking the surface-bound TGF-β on Tregs repressed anti-tumor cytotoxicity regulated by CD8^+^ T cells ex vivo [98]. Thus, a targeted disorder of tumor-enabling signaling within the TME or combining TGF-β antagonists with immuno-mediating therapies could be therapeutic strategies that conquer adverse events related to TGF-β blockade. Additionally, new small-molecule inhibitors of TGFBR1 with controllable toxicity are being developed. Based on the results that TGF-β blockade enhances response to radiotherapy [184], the use of TGF-β-targeted agents combined with chemoradiation could theoretically lead to the approval of TGF-β antagonists for clinical utility. For patients with PC, TGF-β-specific T cells showed significant clinical advantage and prolonged survival after the combined use of radiotherapy and ICIs [203]. Most recently, a comprehensive meta-analysis demonstrated an increase in progression-free survival, OS, and ORR when chemotherapy or radiotherapy was added to TGF-β antagonists in solid tumors [204]. Advancement in understanding the collective activities of TGF-β in specific tumor subtypes and various stages of GI cancer, identification of dependable and precise biomarkers, and evaluation of proper methods to reduce adverse events may provide beneficial insights into the clinical determinations in selecting patients who are sensitive to TGF-β-targeted therapies.

## 10. Conclusions

It is acknowledged that dysregulation of the TGF-β pathway is closely correlated with the progression of various GI cancers. The pleotropic activity of TGF-β is exploited by its capacity to regulate diverse biological processes, including EMT, angiogenesis, immune evasion, and malignant progression at a late stage. Nevertheless, a comprehensive understanding of the molecular mechanisms underlying its different roles in the advancement of GI cancers, as well as identifying reliable biomarkers and TGF-β antagonists exhibiting greater specificity and minimum toxicity, is essential in the development of more useful treatment approaches, possibly in combination with other antitumor agents, to enhance therapeutic efficacies and prolong the survival of patients with GI cancers.

## Figures and Tables

**Figure 1 biomolecules-13-01551-f001:**
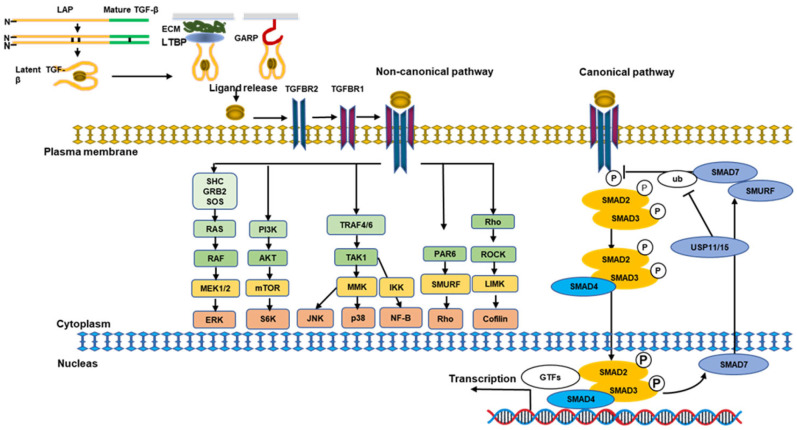
Schematic diagram of transforming growth factor-β (TGF-β) signaling pathways. A latent complex in the collagen comprised an N-terminal peptide with a latency-associated peptide (LAP) and a mature C-terminal fragment. The N-terminal signal peptide is removed, the pro-peptide accumulates into a disulfide-linked homodimer, and latent TGF-β is transformed into homodimeric ligands. This latent TGF-β molecule can create a complex with latent TGF-β-binding protein (LTBP) that mediates its deposition to the extracellular matrix (ECM), mediating the release of active TGF-β via interaction. TGF-β can also be activated by anchoring to glycoprotein-A repetitions predominant (GARP). Activated TGF-β phosphorylates TGF-β type 2 receptor (TGFBR2), which recruits TGF-β type 1 receptor (TGFBR1) and connects to a tetrameric complex composed of TGF receptors and then activates canonical and noncanonical TGF-β pathways. In the case of the canonical SMAD pathway, TGBFR1 recruits and phosphorylates specific SMADs such as SMAD2 and SMAD3, which can form heteromeric complexes with SMAD4. Then, SMAD2/3 and the SMAD4 complex translate into the nucleus to regulate target gene transcription. TGF-β also activates non-canonical pathways when connected to other downstream factors, such as SHC/GRB2/SODS-MAPK pathways, RHO-like GTPase signaling pathways, TRAF4/6, PAR6, and PI3K/AKT pathways.

**Figure 2 biomolecules-13-01551-f002:**
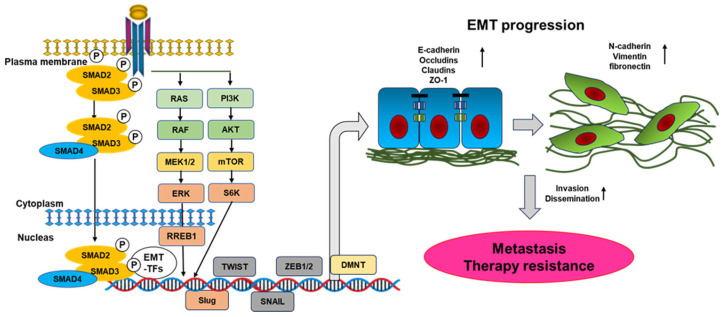
Epithelial–mesenchymal transition (EMT) process. Epithelial cells show apical-basal polarity, and they are connected strongly through tight junctions and adherence junctions, and desmosomes are bound to the extracellular membranes. TGF-β activates the SMAD2/3/4 complex and then regulates EMT-inducing transcription factors such as SNAIL, TWIST, Slug, DMNT, and ZEB. These factors repress epithelial genes and include mesenchymal genes. Expression of these genes leads to cellular alterations, including disassembly of epithelial cell-cell junctions and deficiency of apical-basal cell polarity, where cells become disseminated with the ability for invasion. Activation of non-canonical signaling responses such as MAP kinases and the PI3K/Akt pathway in response to TGF-β has been connected to TGF-β-induced EMT via their mediation of cytoskeleton organization, cell growth, apoptosis, motility, or invasion. Through EMT, the down-regulation of epithelial markers such as E-cadherin, occludins, and claudins is shown, whereas mesenchymal markers such as N-cadherin, vimentin, and fibronectin are up-regulated.

**Figure 3 biomolecules-13-01551-f003:**
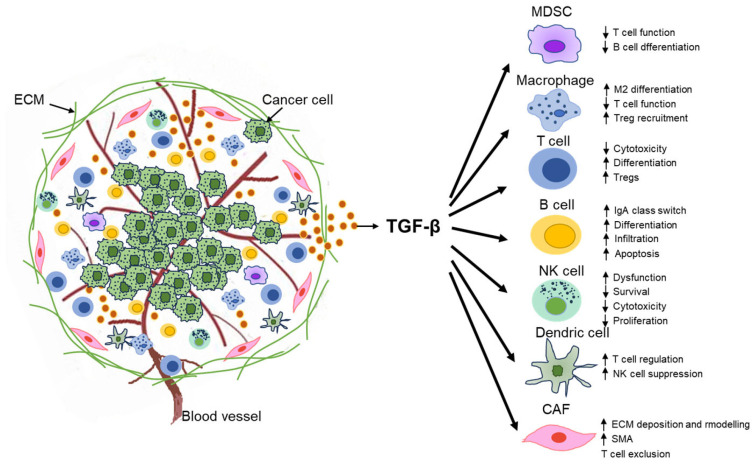
Tumor microenvironment-related cells and extracellular elements as regulators of cancer progression: The schematic illustrates how the interaction between the tumor cells and the surrounding stroma cells involves myeloid-derived suppressor cells (MDSCs), macrophages, T cells, B cells, natural killer (NK) cells, dendritic cells (DCs), and cancer-associated fibroblasts (CAFs). TGF-β modulates cancer immunity. TGF-β suppresses T-helper 1 (Th1), M1, and N1 but facilitates Th2, M2, and N2 differentiation and advancement in the tumor microenvironment. As well, TGF-β inhibits NK and DC functions.

**Table 1 biomolecules-13-01551-t001:** Frequency of genetic alterations in TGF-β pathway-related genes in GI cancers. GI cancer = gastrointestinal cancer; CRC = colorectal cancer; PC = pancreatic cancer; EC = esophageal cancer; GC = gastric cancer; HCC = hepatocellular carcinoma; TGF-β = transforming growth factor-β; Ref = reference; N.A. = not applicable.

Cancer Type	TGF-β 1/2/3	TGFBR1	TGFBR2	SMAD1	SMAD2	SMAD4	Ref
**EC**	Mutation 4% Down-regulated 20%	Mutation 1% Down-regulated 5%	Mutation 1% Down-regulated 7%	Mutation 1% Down-regulated 8%	Mutation 1% Down-regulated 20%	Mutation 6–22% Complete deletion 9.8%	[46,47,48]
**GC**	Mutation 4–13% Up-regulated 22.8%	Mutation 1% Loss 7%	Down-regulated 45%	N.A.	Mutation 0%	Mutation 6.0% Up-regulated 0.2% Complete deletion 3.7% LOH 61%	[48,49,50,51]
**CRC**	Mutation 1% Loss 8% Up-regulated 58.8/26.4/23.5%	Mutation 1% Down-regulated 8%	Mutation 11–30%	Mutation 8.6%	Mutation 2.3–10.3%	Mutation 12–38% Complete deletion 0.9% LOH 61%	[39,48,49,52,53,54]
**PC**	Mutation 1% Loss 20% Up-regulated 47%	Mutation 1–5% Down-regulated 20%	Mutation 4–7% Somatic alteration 4.1%	N.A.	N.A.	Mutation 20.4% Deletion 13% LOH 90%	[49,55]
**HCC**		TGF-β 1/2/3	Mutation 1%	Mutation 3.0%	Mutation 3.0%	Mutation 0.8–6% Complete deletion 0.2%	[48,49]
**Biliary tract** **cancer**	Mutation 1% Down-regulated 4%	N.A.	Mutation 50%	N.A.	N.A.	Mutation 3.9–16%	[48,56,57]

**Table 2 biomolecules-13-01551-t002:** ncRNAs associated with TGF-β signaling in GI cancer for the past 5 years. ncRNAs = non-coding RNAs; TGF-β = transforming growth factor-β; GI cancer = gastrointestinal cancer; Ref = reference; CRC = colorectal cancer; PC = pancreatic cancer; EC = esophageal cancer; GC = gastric cancer; HCC = hepatocellular carcinoma; lncRNA = long non-coding RNA; circRNA = circular RNA; piRNA = PIWI interacting RNA; EMT = epithelial-mesenchymal transition; MSI = microsatelite instability; ICB = immune checkpoint blockade; EndMT = endothelial-to-mesenchymal transition; TAM = tumor-associated macrophage; EV = extracellular vesicles; Down = downregulation; Up = upregulation; N.A. = not applicable.

	Cancer Type	Alterations	Interaction	Comments	Ref
miRNA					
miR-130a-3p	EC	Down	Suppression	TGF-β/miR-130a-3p/SMAD4 pathway mediated by EMT.	[112]
miR-93-5p	EC	Up	Activation	Prompts the migration and inhibits the apoptosis by targeting TGFβR2.	[113]
miR-21-5p	EC	Up	Activation	Activates M2 macrophages and, in turn, EMT through TGF-β/Smad2 signaling.	[114]
miR-181a	EC	Up	Activation	Promotes EMT through the TGF-β/SMAD pathway.	[115]
miR-21	EC	Up	Activation	TGF-βin CTLs may serve as an excellent biomarker.	[116]
miR-577	GC	Up	Activation	TGF-β-miR-577-SDPR axis enhances metastasis via the NF-κB pathway.	[117]
miR-187	GC	Up	Suppression	Suppresses CDDP resistance.	[118]
miR-135b	GC	Up	Activation	Prompts growth and migration by negatively modulating the expression of TGFBR2.	[119]
miR-130a-3p	GC	Up	Activation	TGF-β1/SMAD3 signaling via miR-130a-3p increased proliferation.	[120]
miR-106a	GC	Up	Activation	Cancer-derived exosomes transport miR-106a to peritoneal mesothelial cells.	[121]
miR-200c	GC	Up	Suppression	Negatively associated with EMT, angiogenesis, and hypoxia.	[122]
miR-875-5p	GC	Up	Suppression	Downregulates USF2, leading to repression of TGF-β signaling.	[123]
miR-31-3p	CRC	Up	Activation	Positively related to TGFBR2 deficiency in MSI CRC.	[124]
miR-4666-3p and miR-329	CRC	Down	Suppression	Regulate CRC stemness.	[125]
miR-186-5p	CRC	Up	Suppression	Suppresses cell cycle via inhibition of SMAD6/7.	[126]
miRNA-146b, miRNA-155, and miRNA-22	CRC	N.A.	N.A.	Three miRNAs identify SMAD2, SMAD4, and TGFBR2 as companions for ICB.	[127]
miR-200	CRC	Up	Activation	EV-encapsulated miR-200 regulates tumor-stroma crosstalk in CRC.	[128]
miR-20a-5p	CRC	Down	Suppression	Negatively correlated with LAMTOR5-AS and TGFBR2.	[129]
miR-552	CRC	Down	Suppression	Suppresses 5-FU resistance.	[130]
miR-495-3p	CRC	Up	Suppression	Negatively mediates TGFβR1, TGFβR2, and SMAD4 genes.	[131]
miR-501-3p	PC	Up	Activation	M2 macrophage-derived exosomal miR-501-3p suppresses the TGFBR3 gene pathway.	[111]
miR-21	PC	Up	Activation	M2 macrophage-derived exosomal miR-501-3p suppresses the TGFBR3 gene pathway.	[111]
miR-622,	PC	Up	Suppression	Targets HULC and suppresses invasion by blocking EMT signaling through EV transfer.	[132]
miR-492	PC	Up	Activation	Promotes malignant behavior through the NR2C1-TGF-β/Smad3 pathway.	[109]
miR-133a	HCC	Up	Suppression	Repressed malignant behavior of HCC via TGF-β/Smad3 signaling.	[133]
miR-362-3p	HCC	Up	Activation	Regulates EMT by modulating CD82 and TAM.	[134]
miR-494	HCC	Up	Activation	Regulates EndMT via SIRT3/TGF-β/SMAD signaling.	[135]
miR-141	HCC	Up	Suppression	Blocks growth and invasion by directly downregulating TGFβR1.	[136]
miR-140-5p	HCC	Down	Suppression	Serves as an EMT inhibitor.	[137]
miR-17-5p	HCC	Down	Suppression	Inhibits TGFβR2 expression and EMT.	[138]
miR-324-5p	Gallbladder cancer	Down	Suppression	Targets TGFB2 expression to inhibit GBC cell metastatic behaviors.	[139]
**lnc RNAs**					
NCK1-AS1	EC	Up	Activation	Prompts migration and invasion.	[140]
FAM83H-AS1	EC	Up	Activation	Induced by TGF-β and increased migration and invasion.	[141]
LINC00665	GC	Up	Activation	Increases growth, invasion, and metastasis.	[142]
SGO1-AS1	GC	Up	Suppression	Suppresses the TGF-β pathway and impaired gastric carcinoma metastasis.	[143]
Nr2F1-AS1	GC	Up	Activation	Promotes progression by regulating the miR-29a-3p/VAMP7 axis induced by EMT	[144]
MBNL1-AS1	GC	Down	Suppression	Regulates the miR-424-5p/Smad7 axis and prompts TGF-β/EMT pathways.	[106]
XLOC_004787	GC	Up	Activation	Enhances growth and migration via TGF-signaling and blockade of mir-203a-3p.	[145]
LOC646329	CRC	Up	Activation	Suppresses CRC development via sponging miR-29b and modulates TGFB signaling.	[146]
SNHG6	CRC	Up	Activation	Stimulates TGF-β/Smad signaling via targeting UPF1 and inducing EMT.	[147]
CASC9	CRC	Up	Activation	Predicts the prognostis for patients with CRC.	[148]
MIR-22HG	CRC	Down	Suppression	Related to CD8A and induced T cell infiltration.	[149]
TUG1	CRC	Up	Activation	TUG1/TWIST1/EMT signaling is associated with CRC metastasis prompted by TGF-β.	[150]
CTBP1-AS2	CRC	Up	Activation	Stimulates the TGF-β/SMAD2/3 pathway by blocking miR-93-5p.	[151]
LINC00941	CRC	Up	Activation	Stimulates the TGF-β/SMAD2/3 axis.	[104]
VPS9D1-AS1	CRC	Up	Activation	VPS9D1-AS1/TGF-β signaling enhances tumor proliferation and immune evasion.	[152]
PVT1	PC	Up	Activation	Serves as an oncogene via EMT via the TGF-β/Smad pathway.	[153]
Linc00462	PC	Up	Activation	Enhances TGFBR1/2 expression and activates the SMAD2/3 pathway.	[108]
MIR100HG	PC	Up	Activation	Regulates TGFβ signaling through TGFβ1 production.	[154]
MIR31HG	PC	Up	Activation	*MIR31HG* inhibits TGFβ-induced EMT and cancer cell migration.	[155]
SBF2-AS1	HCC	Up	Activation	Regulates TGFBR1 via sponging miR-140-5p.	[105]
UCA1	HCC	Up	Activation	TGF-β1 prompts the growth of HCC via the upregulation of UCA1 and HXK2.	[156]
SNAI3-AS1	HCC	Up	Activation	Regulates UPF1 and activates the TGF-β/Smad pathway, leading to HCC progression.	[157]
NORAD	HCC	Up	Activation	Regulates the TGF-β pathway to stimulate HCC development by miR-202-5p.	[158]
MEG3	HCC	Up	Suppression	Increase growth, migration, and invasion.	[159]
LncRNA34a	HCC	Up	Activation	LncRNA34a is positively associated with bone metastasis.	[160]
ELIT-1	HCC	Up	Activation	Positively regulates TGFβ/Smad3 signaling and EMT.	[161]
SLC7A11-AS1	HCC	Up	Activation	SLC7A11-AS1 and hsa_circ_0006123 are implicated in the EMT induced by TGF-β.	[162]
LINC01278	HCC	Up	Activation	β-catenin/TCF-4-LINC01278-miR-1258-Smad2/3 feedback loop is associated with HCC metastasis.	[163]
B3GALT5-AS1	HCC	Up	Activation	Suppresses CD4 T cell invasion through TGF-β signaling.	[164]
LINC00261	HCC	Up	Suppression	Represses EMT and stem-like change by blocking TGF-β1/SMAD3 signaling.	[165]
PRR34-AS1	HCC	Up	Activation	Prompts the exosome secretion of VEGF and TGF-β.	[166]
**circ RNAs**					
circCACTIN	GC	Up	Activation	The circCACTIN/miR-331-3p/TGFBR1 axis promotes migration, invasion, and EMT.	[167]
circTHBS1	GC	Up	Activation	Promotes malignant behavior and EMT.	[168]
circ_0006089	GC	Up	Activation	Prompts glycolysis and angiogenesis via the miR-361-3p/TGFB1 pathway.	[169]
Circ-E-Cad	GC	Up	Activation	Regulates growth, migration, and EMT via PI3K/AKT signaling.	[170]
circPACRGL	CRC	Up	Activation	Exosomal circPACRGL enhanced malignant behavior via the miR-142-3p/miR-506-3p-TGF-β1 axis.	[171]
circPTEN1	CRC	Down	Suppression	Disordered SMAD4 interaction with SMAD2/associated with EMT upon TGF-β activation.	[172]
circANAPC7	PC	Down	Suppression	Represses tumor proliferation via the CREB-miR-373-PHLPP2 axis and TGF-β down-regulation.	[173]
circSPECC1	HCC	Up	Activation	Mediates TGFβ2 and autophagy via miR-33a under oxidative stress.	[174]
circGFRA1	HCC	Up	Activation	Regulates the miR-498/NAP1L3 axis, which led to HCC development.	[107]

**Table 3 biomolecules-13-01551-t003:** Clinical trials of pharmacological strategies targeting TGF-β for GI cancer. GI cancer = gastrointestinal cancer; TGF-β = transforming growth factor-β; TGFBR1 = TGF-β2 type receptor; TGFBR2 = TGF-β2 type receptor; PD-L1 = programmed death-1 ligand; CRC = colorectal cancer; PC = pancreatic cancer; EC = esophageal cancer; GC = gastric cancer; GEJC = gastroesophageal junction cancer; HCC = hepatocellular carcinoma; CCC = cholangiocellular carcinoma; COX = cyclooxygenase.

Strategy	Agents	Target	Cancer Type	Identifier	Status	Results	Comments
Antibody							
	SAR439459	TGF-β 1/2/3	CRC, HCC	NCT03192345	Ib	Termninated	Monotherapy +Cemiplimab
	NIS793	TGF-β 1/2/3	PC	NCT04390763	II	Active	Monotherapy +spartalizumab
	NIS793	TGF-β 1/2/3	PC	NCT04935359	III	Active, not-recruiting	Gemcitabine and nab-paclitaxel ± NIS793
	NIS793	TGF-β 1/2/3	PC	NCT05417386	I	Recruiting	+FOLFIRINOX
	NIS793	TGF-β 1/2/3	CRC	NCT04952753	II	Active, not-recruiting	+bevacizumab with mFOLFOX6 or FOLFIRI
	ABBV151	GARP:TGF-β1	PC HCC	NCT03821935	I	Recruiting	Monotherapy +ABBV151
	GT90001	ALK-1	GC, EC, HCC	NCT04984668	I/II	Recruiting	+KN046
	ALK-1	HCC	NCT05178043	II	Recruiting	+Nivolumab	GT90001
**Ligand trap**							
	M7824	TGF-β/PD-L1	CCC/gallbladder cancer	NCT03833661	II	Active	Monotherepy
	M7824	TGF-β/PD-L1	Biliary tract cancer	NCT04066491	II	Recruiting	+gemcitabine and cisplatin
	M7824	TGF-β/PD-L1	PC	NCT03451773	II	Completed	+gemcitabine
	M7824	TGF-β/PD-L1	PC	NCT04327986	I	Terminated	+M9241 and radiotherapy
	M7824	TGF-β/PD-L1	CRC	NCT03436563	II	Active, not recruiting	Monotherapy
	M7824	TGF-β/PD-L1	Small bowel cancer, CRC	NCT04491955	II	Active	+N-803, M9241, and CV301
	M7824	TGF-β/PD-L1	Small bowel cancer, CRC	NCT04708470	I/II	Recruiting	Entinostat and NHS-IL12 ± M7824
	M7824	TGF-β/PD-L1	EC	NCT04595149	II	Recruiting	+paclitaxel, carboplatin, and radiotherapy
	M7824	TGF-β/PD-L1	Intrahepatic cholangiocarcinoma	NCT04708067	I	Recruiting	+Hypofractionated radiation
**Small-** **molecule inhibitors**							
	Galunisertib	TGFBR1	HCC	NCT02906397	I	Active	+radiotherapy
	Galunisertib	TGFBR1	Rectal cancer	NCT02688712	II	Active, not recruiting	+fluorouracil/capecitabine and radiotherapy
	Galunisertib	TGFBR1	PC	NCT02734160	II	Terminated	+durvalumab
	Galunisertib	TGFBR1	HCC	NCT01246986	II	Active	+sorafenib/ramucirumab
	Galunisertib	TGFBR1	PC	NCT01373164	II	Completed	+gemcitabine
	Vactosertib	TGFBR1	HCC	NCT02160106	I	Completed	Monotherapy
	Vactosertib	TGFBR1	CRC	NCT03844750	I	Recruiting	+pembrolizumab
	Vactosertib	TGFBR1	GC	NCT03698825	II	Recruiting	+paclitaxel
	Vactosertib	TGFBR1	GC	NCT04656002	II	Not yet recruiting	+paclitaxel and ramucirumab
	Vactosertib	TGFBR1	GC	NCT04893252	II	Recruiting	+durvalumab
	Vactosertib	TGFBR1	PC	NCT04258072	II	Active	+irinotecan, fluorouracil and leucovorin
	LY3200882	TGFBR1	CRC	NCT04031872	II	Active	+capecitabine
	PF06952229	TGFBR1 (ALK1)	CRC	NCT02116894	I	Completed	+regorafenib
	PF06952229	TGFBR1 (ALK1)	PC, CRC, HCC	NCT03685591	I	Recruiting	Monotherapy
	GFH018	TGFBR1	PC, EC, CRC, HCC, Biliary tract cancer	NCT04914286	I/II	Recruiting	+Toripalimab
	TEW-7197	ALK4/5	PC	NCT03666832	I/II	Recruiting	+FOLFOX
	SHR-1701	TGFBR2/PD-L1	CRC	NCT04856787	II/III	Recruiting	+BP102 and capecitabine/oxaliplatin
	SHR-1701	TGFBR2/PD-L1	Rectal cancer	NCT05300269	II	Recruiting	+capecitabine/oxaliplatin
	SHR-1701	TGFBR2/PD-L1	GC	NCT05149807	I	Enrolling by invitation	Monotherapy
	SHR-1701	TGFBR2PD-L1	HCC	NCT04679038	I/II	Recruiting	+Famitinib
	SHR-1701	TGFBR2/PD-L1	GC/GEJC	NCT04950322	III	Recruiting	+capecitabine/oxaliplatin
**Antisense oligonucleotide**							
	STP705	TGF-β/COX-2	HCC, CCC	NCT04676633	I	Active, not recruiting	Monotherapy
	Trabedersen	TGF-β mRNA/TGFBR2	CRC	NCT00844064	I	Completed	Monotherapy
**Vaccine**							
	vigil	TGFβI/TGFβII	Liver cancer	NCT01061840	I	Completed	Monotherapy
	STP705	TGF-β/COX-2	HCC, CCC	NCT04676633	I	Active, not recruiting	Monotherapy

## Data Availability

The data presented in this study are available in this article.

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
