# Peer review of "The Role of the Transforming Growth Factor-β Signaling Pathway in Gastrointestinal Cancers"

_biomolecules, 2023, doi:10.3390/biom13101551_

Round 1
Reviewer 1 Report
Manuscript ID: biomolecules-2650164
Type of manuscript: Review
Title: The Role of Transforming Growth Factor-β Signaling Pathway 2 in Gastrointestinal Cancers
Journal: BIOMOLECULES
Growing evidences have been shown the TGF-β is a factor that promotes the malignant transformation of cancer and in particular the progression and evolution of cancer cells. Interestingly, in this review, the authors have described a complex overview of the mechanisms and functions of TGF-β signaling and therefore the therapeutic potentials of targeting the TGF-β signaling .
The paper is well organized and well written and represents a contribution very important in the field of the TGF-β signaling and cancer. I personally recommend the publication of this review.
I have only a suggestion: the authors could expand and discuss the role of fibrosis in the evolution of cancer linked to TGF-β signaling.
Author Response
We appreciate the reviewer’s comments. According to the reviewer’s suggestion, we added the role of fibrosis in the progression of cancer linked to TGF-β signalling (page 8, lines 302-315).

Reviewer 2 Report
The review article describes the role of transforming growth factor-b signaling pathway in gastrointestinal cancers. The article is interesting. Nevertheless, it should be improved before publication. The flow of the article and presentation of the data from original articles should be improved. More tables should be included to better summarize the content of chapters. Special focus should be on table describing non-coding RNAs involved in TGF-b signaling.
Minor editing of English language required.
Author Response
We appreciate the reviewer’s comments. According to the reviewer’s suggestion, we added a new chapter “Role of non-coding RNAs involved in TGF-β signaling” and new Table 2 in the revised manuscript.

Reviewer 3 Report
There are many studies regarding TGF-beta indicating an interest in this molecule in tumor progression and therapeutic strategies as a target. The authors have demonstrated a good understanding of the current state of knowledge and a strong summarization ability, therefore I consider this review to be accurate and of interest to the scientific community. For more new data I suggest including the references following reports with a comment in the text.
1. Hosseini, R.; Hosseinzadeh, N.; Asef-Kabiri, L.; Akbari, A.; Ghezelbash, B.; Sarvnaz, H.; Akbari, M.E. Small Extracellular Vesicle TGF-β in Cancer Progression and Immune Evasion. Cancer Gene Therapy 2023, doi:10.1038/s41417-023-00638-7.
2. Rahavi, H.; Alizadeh-Navaei, R.; Tehrani, M. Efficacy of Therapies Targeting TGF-β in Solid Tumors: A Systematic Review and Meta-Analysis of Clinical Trials. Immunotherapy 2023, 15, 283–292, doi:10.2217/imt-2022-0079.
3. Mortensen, R.E.J.; Holmström, M.O.; Lisle, T.L.; Hasselby, J.P.; Willemoe, G.L.; Met, Ö.; Marie Svane, I.; Johansen, J.; Nielsen, D.L.; Chen, I.M.; et al. Pre-Existing TGF-β-Specific T-Cell Immunity in Patients with Pancreatic Cancer Predicts Survival after Checkpoint Inhibitors Combined with Radiotherapy. Journal for ImmunoTherapy of Cancer 2023, 11, doi:10.1136/jitc-2022-006432.
as they regard immune modalities by T-cell specific, a recent metanalysis of the-b blockade combined with chemo-radio therapies and a review on diagnostics, therapeutic and prognostic importance of TGF-b related-EVs
Author Response
We appreciate the reviewer’s comments. According to the reviewer’s suggestion, we added the references following reports with a comment in the text described below.
Hosseini, R.; Hosseinzadeh, N.; Asef-Kabiri, L.; Akbari, A.; Ghezelbash, B.; Sarvnaz, H.; Akbari, M.E. Small Extracellular Vesicle TGF-β in Cancer Progression and Immune Evasion. Cancer Gene Therapy 2023, doi:10.1038/s41417-023-00638-7.
→Page 20, lines 530-532. Reference number: 199.
Rahavi, H.; Alizadeh-Navaei, R.; Tehrani, M. Efficacy of Therapies Targeting TGF-β in Solid Tumors: A Systematic Review and Meta-Analysis of Clinical Trials. Immunotherapy 2023, 15, 283–292, doi:10.2217/imt-2022-0079.
→ Page 21, lines 549-551. Reference number: 204.
Mortensen, R.E.J.; Holmström, M.O.; Lisle, T.L.; Hasselby, J.P.; Willemoe, G.L.; Met, Ö.; Marie Svane, I.; Johansen, J.; Nielsen, D.L.; Chen, I.M.; et al. Pre-Existing TGF-β-Specific T-Cell Immunity in Patients with Pancreatic Cancer Predicts Survival after Checkpoint Inhibitors Combined with Radiotherapy. Journal for ImmunoTherapy of Cancer 2023, 11, doi:10.1136/jitc-2022-006432.
→ Page 21, lines 547-549. Reference number: 203.
